# Parents' perception on cause of malaria and their malaria prevention experience among school-aged children in Kutcha district, Southern Ethiopia; qualitative study

Zerihun Zerdo [1,2]*, Jean-Pierre Van Geertruyden[2], Fekadu Massebo[3], Gelila Biresaw[1], Misgun Shewangizawu[4], Abayneh Tunje[4], Yilma Chisha[4], Tsegaye Yohanes[1], Hilde Bastiaens [5‡], Sibyl Anthierens[5‡]

1 Department of Medical Laboratory Science, College of Medicine and Health Sciences, Arba Minch University, Arba Minch, Ethiopia, 2 Global Health Institute, Antwerp University, Antwerp, Belgium, 3 Department of Biology, College of Natural Sciences, Arba Minch University, Arba Minch, Ethiopia, 4 Department of public health, College of Medicine and Health Sciences, Arba Minch University, Arba Minch, Ethiopia, 5 Department of Family Medicine and Population Health, University of Antwerp, Antwerp, Belgium

‡ These authors are joint senior authors on this work.
* zedozerihun@gmail.com

**Data Availability Statement:** All relevant data are within the manuscript.

## Abstract

### Introduction

School-aged children become a highly vulnerable group for malaria, yet they are less likely to use malaria prevention interventions. Previous studies exploring perception on cause of malaria mainly focused on pregnant mothers or parents of children under age five years. Exploring parent's perception on cause of malaria and their experiences on the prevention of malaria and associated challenges among school-aged children is important to develop a malaria prevention education package for school-aged children to reduce malaria and malaria related morbidities among school-aged children.

### Methods

A descriptive qualitative study is conducted in Kutcha district by recruiting 19 parents of school-aged children for semi-structured interviews, 6 key informants and 6 focus group discussion which consists of parents, health development army and health extension workers. A semi-structured interview guide is used to guide the interview process. The collected data is analyzed thematically with a focus on the three major areas of concern: perceived cause of malaria, experience of malaria prevention and challenges of bed net use for prevention of malaria.

### Results

Five causes of malaria were identified, namely hunger, mosquito bite, exposure to hot sunshine, poor sanitation and hygiene and eating some sweet foods and unripe maize. Participants perceived that eating sweet foods and unripe maize lead to enlargement of the spleen

**Funding:** The authors received no specific funding for this work.

**Competing interests:** The authors have declared that no competing interests exist.

that ends in malaria while poor hygiene and sanitation leads to either development of the ova of mosquito and the landing of the housefly to contaminate food for consumption. The experiences of malaria prevention were largely influenced by their perceived cause of malaria. The malaria prevention measures undertaken by parents were vectors control measures, homemade herbal remedies and restricting children from eating sweet foods. The challenges of malaria prevention by using bed nets were related to a negative attitude, sleeping behaviors of children; use of bed nets for unintended purposes, shortage of bed nets and delays in the distribution of bed nets.

## Conclusion

There were misconceptions about the cause of malaria and associated experiences of malaria prevention. Control of malaria among school-aged children need health education targeting the challenges and correcting identified misconceptions by parents in Kutcha district and in other similar settings.

## Introduction

In the presence of effective prevention and treatment strategies, malaria remains the major cause of morbidity and mortality in Tropical and subtropical countries in the world. According to the World Health Organization report (WHO) 2018, an estimated 219 million malaria cases in the globe were found, but 92% of the cases were in the WHO African region [1]. Malaria is caused by protozoan parasites of the genus Plasmodium and transmitted through the bite of the female anopheles mosquito. There are five known species of Plasmodium that cause malaria in humans. These are *Plasmodium (P).falciparum*, *P.vivax*, *P.ovale*, *P.malariae and P.knowlesi* that is zoonotic parasite and did not exist in Africa [2]. All human Plasmodium species causing malaria were found in Ethiopia but *P. falciparum* and *P.vivax* were responsible for almost 60% and 40% of malaria related cases, respectively [3, 4].

In the last two decades, malaria was reduced due to intensified malaria control strategies by different actors. The global incidence of malaria was reduced by 41% between 2000 and 2015. The dramatic decline in Sub-Saharan Africa, where almost all malaria related deaths occur was observed after the introduction of long-lasting insecticide treated bed nets in 2010 [5]. However, the overall global malaria incidence remained stable for the last three years and making off track to achieve the 40% reduction in annual malaria incidence and death rate milestones of global technical strategy for malaria by 2020. In some of the WHO regions, there was no significant change; there was an increase in South America, but the decrease was promising in Africa [1]. The average annual incidence of malaria in Ethiopia was reduced by 22.9% every year between 2010 and 2016. Moreover, the reported national malaria cases of children age less than five years decreased by 59.3% between the years 2013 and 2016 [4].

As indicated above, the intensity of exposure to Plasmodium species in young children was decreased. This resulted in delayed development of naturally acquired protective immunity and made school-aged children (SAC) to become highly susceptible to malaria [6, 7]. Malaria became responsible for 50% of SAC morbidity and mortality in SSA [8]. In the most recent national malaria indicator survey conducted in Ethiopia, the prevalence of malaria was even higher among SAC than children age < 5 years and pregnant mothers [9]. In other countries in Africa, in a similar age group, both clinical malaria and subclinical malaria parasitemia were high. Up to 76.4% of SAC were infected by Plasmodium parasites in Tanzania [10, 11]. The

corresponding numbers in Malawi [12], Rewanda [13] and Ethiopia [14] were 31, 22.4 and 14.5 respectively. Above all, SAC become the major source of infection to the mosquito vector and thus, serve as source of infection to other susceptible individuals [12].

The consequence of malaria among SAC is multidimensional. Malaria among SAC has a negative impact on the behavior of the child, impairs physical as well as cognitive development and reduces executive capabilities in the school performance which partly emanates from school absenteeism. If left untreated, malaria can be complicated and results in death and morbidities such as anemia and malnutrition [13, 15–19]. For parents of children suffering from malaria, it leads to psychological trauma, distorts social relations, lowers the economic development as a result of treatment costs and loss of time to work and it even can become the cause of a separation of the parents of the child when it becomes impossible to afford treatment and associated costs [19–21]. Fortunately, malaria can be prevented by using proven preventive strategies and can be treated effectively using available anti-malarial drugs.

The two major strategies for the prevention and control as well as for subsequent elimination of malaria are early diagnosis and treatment of malaria cases and vector control. One of the two powerful vector control strategy was the consistent utilization of bed nets [1, 3, 22]. Despite their increased susceptibility to malaria and apparent reports of high burden, SAC were the least benefited from malaria intervention strategies [23, 24]. The involvement of wider community and multi-sectorial collaborations including the school sector is important for the universal coverage of major prevention strategies [25]. The acceptability and participation of these stakeholders depend on their level of awareness [26]. For example, utilization of bed nets by these children is dependent on the perception of parents or caregivers on malaria, attitude towards malaria prevention strategies as well as perceived susceptibility of their children to malaria [21, 24].

Previous studies on the perception of malaria and treatment seeking behavior are focused on either pregnant mother, mother of children aged less than five years or household heads [26–29]. Less is known about the perception of parents of SAC on the cause of malaria and their experience on the control of malaria. The perception of parents on the cause of malaria are diverse and context dependent: some consider it as supernatural [30] while others associate it with poor environmental and personal hygiene, consumption of contaminated water or food, exposure to different weather condition to mention few among others [28, 29, 31]. However, the existing information was not sufficient to develop malaria prevention education that might help in controlling malaria in Ethiopia and other similar contexts. In this study, we have explored perceptions of parents of SAC on cause of malaria, experiences with malaria prevention and challenges of malaria prevention using bed nets in Kutcha district. The finding of this study will be used to develop malaria prevention education package addressing current perceptions and experiences of parents on the cause of malaria and the challenges they describe an effective and consistent utilization of a bed net.

## Methods and materials

### Study area

This study is conducted in Kutcha district in Gamo Zone (the next higher administrative unit next to regional states). Kutcha district is located at about 180 km in the northwestern part of Arba Minch town, capital of Gamo zone. Arba Minch is located at 505 km away from Addis Ababa (the capital of Ethiopia) in South Nations, Nationalities and Peoples Regional (SNNPR) state. According to the 2007 national census conducted by central statistical agency, a total of 149287 people were living in the district. More than 75% of the geographical area of the district is endemic for malaria transmission [32].

## Study design and sampling

A descriptive qualitative study design [33] is used to explore the perception on cause of malaria, lived experience of parents of SAC to prevent malaria and the challenges of malaria prevention in the study district.

Participant triangulation was employed and both parents and key informants were involved. The participants in individual interview were purposively sampled based on a maximum variation sampling strategy with the inclusion criteria of having at least one child aged between5-14 and permanently living in malaria transmission settings in the district. Those included in the interview were from a diverse population category with respect to economic class, gender, age, residence area, kebele leaders, religious leaders, kebele administration officials from different clusters of the malaria transmission area. Homogenous members were involved in each FGD: 1 male parent, 2 female parents, 1 male health development army (HDA), 1 female HDA and 1 health extension workers (HEW). The participants for FGD are selected purposefully with maximum variation and different role players in the health system. HEWs are government paid females trained on the health extension program. There are two HEWs in each kebele (the lower administrative unit in Ethiopia) working as static in Health posts and provide outreach activities within their kebele. HDA are leaders of at least forty households and work jointly with the HEWs.

## Method of data collection

Different data collection techniques were employed: individual interviews, observation, focus group discussions and key informant interviews in order to triangulate the findings and to explore the topic from different angles in order to provide rich data. The topic guides used during the semi-structured interviews are presented in Tables 1 and 2 below.

A semi-structure topic guide was developed focusing on three big themes: parents perceived cause of malaria, their malaria prevention experiences and challenges of malaria prevention through the use of bed nets. These big themes are selected because they determine effective utilization of malaria prevention interventions and consequent burden of malaria among SAC. The individual interviews took place at the participants' homes; direct observation was made to see the condition of the bed nets and its use for unintended purposes.

The data collection process is undertaken in the trusting environment of the participant or under shade surrounding parents and at their office in case of key informants. The semi-structured interview was conducted by two members of the research team who have been trained in qualitative research. FGDs are also conducted in a trusting environment such as at residence

**Table 1. Topic guide used for the semi structured interviews with individual and FGDs.**

| S. NO | Semi-structured core and prompt questions used |
|---|---|
| 1. | What is the common cause of illness in your area? |
| 2. | What do you do if your school-aged child is suffering from febrile illness |
| 3. | What is your perception about the cause of malaria? |
| 4. | How do you prevent your school-aged children from malaria? |
| 5. | What is the purpose of bed net? |
| | For how long bed net is protective from the bite of mosquitoe? |
| | For what other purpose you use the bed net during the time interval in which you perceive it is protective from mosquitoe bite |
| | For what other purposes you use the bed net after the time interval in which you perceive it is protective from mosquitoe bite? |

**Table 2. Topic guide used for the semi structured interviews with key informants, health development army and HEW.**

| S. NO | Semi-structured core and prompt questions used |
|---|---|
| 1. | When school-aged children become fibril, what did people do to alleviate the illness of the child in your community? |
| 2. | When people suspect that their school-aged children were febrile due to malaria, what did they do for their children? |
| 3. | How can people protect their school-aged children from malaria? |
| 4. | What is the purpose of insecticide treated bed nets in your community? |
| 5. | When did school-aged children consistently use bed nets in your community? |
| 6. | How long do you believe that insecticide treated bed nets are protective? |
| 7. | What are the challenges of bed net utilization bed net utilization by the SAC? |
| 8. | For what other purposes do people use insecticide treated bed nets during the time interval in which the bed net is protective for malaria? |
| 9. | For what other purposes do the local people use insecticide treated bed nets after the time interval in which they believe that it is no more protective for malaria? |

of respected people in the area, classrooms in school and in the hospital with the health extension workers. The FGD is accompanied by a coffee ceremony prepared by the research team and volunteers from the area. One of the two data collectors is trained in health education and promotion and the second interviewer is expert of Tropical and Infectious Diseases. The interviews are audio recorded and notes are taken during the data collection process. The interviews were undertaken in Gamogna (local language) and Amharic (commonly spoken language all over Ethiopia) depending on the preference of the participant in the study. Both interviewers involved in the study were fluent in both local languages.

## Ethical consideration

This research is part of a larger cluster randomized controlled trial entitled 'effect of school-based bed net distribution and malaria prevention education on malaria and anemia among SAC in Kutcha district'. This study is approved by verbal consent procedure to be followed by the Institutional Research Ethics review Board (IRB) in College of Medicine and Health Sciences, Arba Minch University with the reference number of ሕጣባ/11275/21. Official permission letter to conduct the research is submitted to the Zonal health department and the district health office. Verbal consent is obtained from each participant after detailed information on purpose, procedure, benefit or harms and autonomy to participate is given. Verbal consent is preferred because majority of the residents in the study area were illiterate, no biological specimen is collected or sensitive issue is discussed. The verbal consent obtained from each participant in the study is documented in college of Medicine and Health Sciences at Arba Minch University. The individual whose image is found in Fig 2 in this manuscript has given written informed consent (as outlined in PLOS consent form) to publish these case details.

## Data analysis

After each interview, there was a debriefing session with the two researchers in order to adapt the topic guide when necessary to improve data collection. The digital audio material is transcribed by one of the two interviewers in verbatim. The transcribed data were read once again and the researchers also listened repeatedly to the audio material in order to deeply immerse themselves into the data. The data were analyzed thematically by one of the researchers (also interviewer) trained in qualitative research methods, but always keeping the main focus of

concern in mind, namely perceived cause of malaria, experience of malaria prevention and challenges of malaria prevention. These areas of concern were used as a guiding frame. First the initial transcripts were coded line by line to unravel the data. After 3 interviews these open codes were segregated based on their similarities in sub-themes. These sub-themes are further grouped and refined to form big themes and discussed in the research team. At the different stages the codes, subthemes and themes were discussed with the 2nd interviewer.

## Result

### Demographic characteristics

A total of 75 participants were included in this study. The mean age of the participants was 38.3 years with the standard deviation of 1.1 years. The occupational status of participants involved in the study was presented in the Fig 1 below. The occupations of participants in the other group in Fig 1 were merchant, kebele leader, church leader, weaver, potter, and school head master and health professionals.

### Findings

**Perceived cause of malaria.** Malaria is perceived by the participants as the major health problem of SAC living in Kutcha district. The perceived causes of malaria were diverse and one participant had more than one perceived cause in most cases. Perceptions about the cause of malaria ranged from participants who did not have any idea to interviewees naming the exact cause and its transmission mechanism. The perceived causes of malaria among SAC could be grouped into five sub themes: hunger, exposure to hot sunshine, poor hygiene and sanitation, mosquito bite and ingestion of some types of food.

**Hunger as cause of malaria.** Almost all individuals, both in the individual semi-structured interview and FGD perceive hunger as the major cause of malaria. According to participants, malaria is caused by failure of the children to take adequate amounts of food, fail to take food at the right time or eating maize continuously without mixing with other food types rich to help the production of blood.

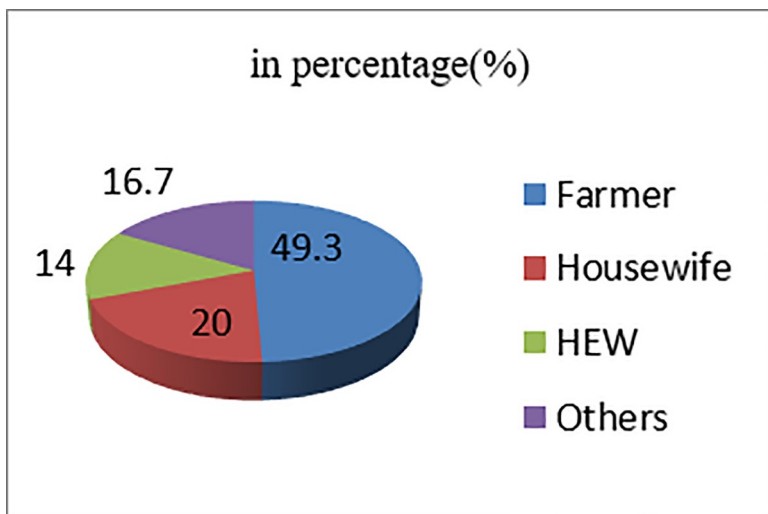

**Fig 1. Occupational status of participants involved in the study in Kutcha district, Southern Ethiopia, 2018.**

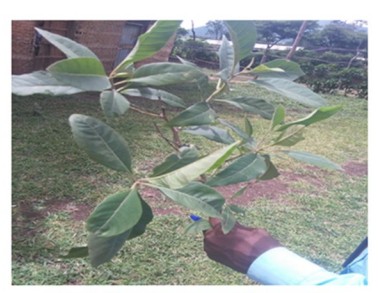
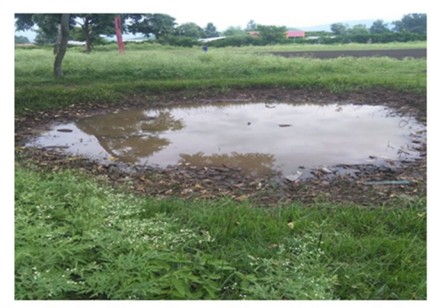
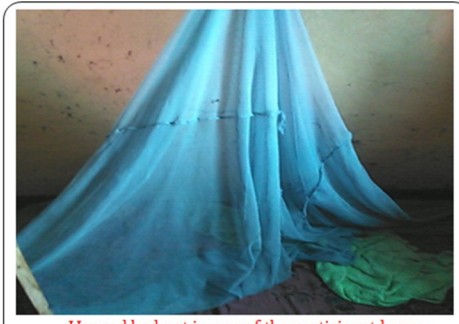
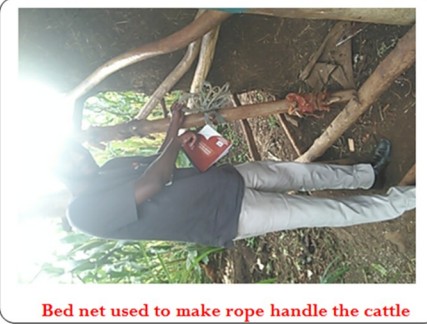
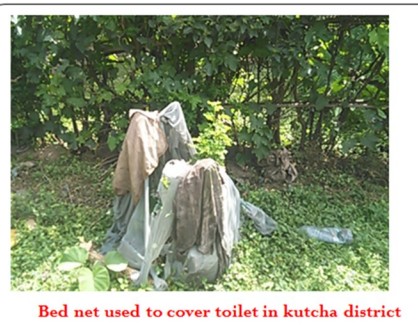

**Fig 2. Pictures of gray leaf, mosquitoe breading site, hanged bed net, misuse of bed net as rope and toilet cover from left to right in this fig.**

*"If a mosquito bites an individual who is in shortage of blood that resulted from hungry, the person bitten by the mosquito will develop malaria. . . Malaria hides itself in the blood and it is triggered to disease state if an individual gets hungry"*-[FGD with mothers].

*"When the climatic condition is very hot and draught is there, malaria is very high as it was scattered down from the sky"*- [FGD with male HDA].

When mosquito bites and hunger were listed as perceived causes of malaria by individual semi-structured interviews and FGD, comparison was made. In both cases, most participants perceived that hunger was the cause of malaria, rather than mosquito bites

*"Is it hunger or mosquito bite that more causes malaria?"*- [interviewer]

*"Malaria is due to bite of mosquito, exposure of children to hot environmental condition and being hunger, but it is hunger that causes more malaria than bite of mosquito"*- [Mother].

**Some food types as cause of malaria.** The other perceived cause of malaria was ingestion of sweet food types such as banana, sugarcane, candy, mango and unripe maize. Participants describe that ingestion of these food types leads to enlargement of spleen, the enlarged spleen is followed by fever which further progresses to malaria. These food types are mentioned in key informants' interview, individual interview and focus group discussion.

*"Because of fear of malaria, I do not buy Asmera type of banana from the market and do not eat and let my children to eat such type of banana, even when there is an abdominal problem to my SAC, I buy Kenya type of banana to eat but never Asmera"*-[young mother].

"*When children eat unripe maize during its flowering season, spleens of the children enlarge. The enlarged spleen is followed by fever, which further develops into malaria*"-[father].

However, there were disagreements in FGD as the above food types cause malaria or not.

"*There is no problem with eating any type of food but the children should eat adequate amount at the right time to become free from malaria*" [FGD with fathers].

**Mosquito as cause of malaria.** Some interviewees clearly indicated that Plasmodium is the causative agent and mosquito is involved in the transmission of the parasite from malaria patient to not infected children. Nevertheless, there are also misconceptions as malaria is caused by mosquito bite that takes the causative agent by landing on contaminated water or unhygienic materials in the surrounding. Still some others hesitate on the role of mosquito bite in the transmission of malaria.

"*I am in trouble to accept that malaria is caused by all mosquitoes. If the mosquito bite causes malaria, it will lead to many more diseased individuals. But some say that it is caused by a mosquito that makes sound. I am asking some persons, but their response is like this and I am in trouble to accept*"-[FGD with fathers].

**Poor hygiene and sanitation.** If the environment surrounding their house is not clean, including dirty toilet, common houseflies land on such dirty material and bring toxins from there to the food which would be ingested by the households. Ingestion of the food contaminated by toxin brought to the food by *Musca demostica* (common housefly) is perceived to be cause of malaria.

"As *I have learned from HEW in my kebele, I am keeping the hygiene of environment and my child to protect him from malaria*"-[father].

Another perceived cause of malaria is the ingestion of contaminated water with ova of the mosquito. Participants explained that dirty water may also attract mosquito and mosquito may lay their eggs on it.

"*Fetching drinking water from river, putting that water in the jar for long time, and then drinking that water causes malaria in addition*"–[mother].

"*Drinking water from the river causes typhoid and that typhoid progress to typhus and then to malaria*. . .*When we drink bad water like flowing river here, the spleen is enlarged and when spleen enlarged person become hungry, then malaria starts*. . .*Those who are living and working away from the pipe, they are drinking rivers which are contaminated and that leads to typhoid fever and typhus and that ends up in malaria. Water from the river should not be drunk and if impossible to get other improved water they shall collect the water during the night time and drink that water to be safe from malaria*"-[FGD with male HDA]

**Exposure to hot sunshine/environment.** The other commonly shared perception on the cause of malaria for children is exposure to hot sunshine. One of the female interviewee in the semi-structured interviewees stated its importance in the following way:

"*Walking in the daylight (hot) environment may cause malaria. Thus, we protect our children from exposure to hot environmental condition*"-[Mother].

Farming in the hot sunshine during the daytime is also perceived as cause of malaria even in adults. The importance of exposure to hot sunshine as a cause of malaria and discordance was discussed in FGD.

"*If a farmer goes to his farm and dig his farmland in the heavy sunshine without breakfast, then that person may become diseased. . . but those coming to the district from northern part of Ethiopia were engaged in heavy work in very hot weather but they were not diseased by malaria*"-[FGD fathers]

**Experience of malaria prevention.** Similar to the prevention strategies used for malaria at a local or global level, participants have been using environmental management like draining pooled water (madafenina mafases in local terms for filling puddles and draining pooled water) and larvicidal chemicals (temephos (Abate®) called as abiy chemical by the local community for the vector control and treatment of diseased children in the health facility. The other prevention strategy our participants use for the prevention of malaria among SAC was bed net and others do nothing.

"*What are you doing in the absence of a bed net? Please go and bring the bed net for us*"- [HEW]. "*There is nothing that I do to prevent my child from malaria and we go to the hospital to take injections if the child becomes symptomatic like white colored body*"-[Mother].

There are some enabling factors to use of bed nets. Some state as a bed net protects children sleeping under it from something falling from the roof such as rats and dust. In contrast, there are also some perceived factors hindering the use of bed nets like its warm inside, reaction to the skin when it is new, suffocation and its ability to attract the bed bugs (mixed views).

The household level prevention experience seemed mostly influenced by their perception of the cause of malaria. They do not let children be exposed to hot sunshine, giving an adequate amount of food at the right time, improving the sanitation and hygiene.

"*. . .other than feeding children on time and praying to God, what can I do to prevent my child from malaria?*"-[Mother]

When the purpose of bed net is asked, all participants responded as it is used to prevent from the bite of mosquito but some reduce its importance much less than taking adequate food to avoid hunger by children.

"*If the house type is local thatched one, taking adequate amount of food is enough to prevent malaria and it does not need using bed net. . . If a child plays outside home in hot environment, he will be diseased by malaria, for this reason, I do not let him to play outside*"- [Father].

A 52 year old father indicated the importance of improved hygiene and sanitation as "*teachers and health extension workers are teaching us to wash child's body and we also tell our children to wash their body and wash their hands to prevent them from malaria and thus, we do that. And what other measure we should take?*"- [Mother]

The other prevention measure is using foods which have bitter taste as opposite to that causing malaria. The plants and some food items that are commonly used for the prevention of malaria by the participants were Black cumin, fenugreek, white garlic, juice of gray leaf.

Fortunately, in the compound of one of the interviewees the gray was there and the picture taken was presented below (1st picture in Fg2).

**Challenges of bed net use for malaria prevention.** The challenges of bed net use for the prevention of malaria from SAC were related to the quality of a bed net, the negative attitude towards prevention of malaria, priority given to use bed net and behavior of the children. All categories of involved participants complain that the recent bed net distributed is not strong; it is short and do not remain in the place after it is released at night to cover people sleeping under the bed net.

"*The previous bed net kills house flies, lice, fleas and any insects landing on it, but the current bed net we had is attracting bed bugs*"-[health personnel].

The other challenge was shortage of bed nets during distribution and failure to distribute it on time. As to the report of all categories of participants in this study, the expected distribution of the bed net to the community is in the previous year of the data collection but at the time of data collection also the bed net is not distributed

Based on the interviews with HEW, it seems that local people believe that it is the task of the government and the health extension workers so they do not take responsibility by themselves.

"*If I tell them to hang the bed net today, the next time while am visiting their household, they returned the bed net back to box. . .the residents believe as they are working everything for the benefit of the HEW*"-[HEW].

In some of their observations, the bed nets may remain hanged over the bed, but it is no longer used at night and still others put some of the bed nets for future use.

"*The mother of my husband (grandmother of SAC) has placed the bed net in a box with her clothes because she brought it from the health center and does not allow anyone to touch it*"-[FGD with Mothers]

The other challenge of malaria prevention using bed net utilization was the least priority given to SAC. In some of the households pregnant mother and children under the age of five years were given priority, but in others it is the father that takes the 1st hand since he is the representative of the house. Behavior related challenges for the prevention of children is the sleeping condition and sleeping time at night. Children get tired while playing during the daytime and may sleep anywhere or on the chairs before dinner. The children may be bitten by the mosquito at that place before they are taken to the bed or may be left there overnight. The sleeping time is delayed due to a coffee ceremony at night in rural parts of the area and people may not sleep before midnight, during that time they may acquire a mosquito bite.

The other challenge of malaria control through control of vector using bed nets is its unintended purpose use. A bed net is sometimes used to make a rope to hold domestic animals (the 4th picture in Fig 2 from left to right) installer of good on horses and donkeys, carry grasses and wood by women. It is also used to cover toilets (the last picture in Fig 2 from left to right), as thread to stitch clothes, can be placed at the bottom of barn to protect grains from insects or not to miss any single grain. It is also used by the residents for filtering some local beverages like keneto, used to separate tef from its chaff and collect coffee beans from the garden, to separate maize from its stem. Bed nets are used as curtains, bed sheets and head scarfs for female in certain households.

"*We are in fear that what will happen in the future if we tell you everything... We use bed nets to cover the toilet, separating seeds from the stem, after thoroughly washing we use it for filtering kineto, coffee and milk during the separation process of milk from the butter. Those who are unable to purchase clothes can use it as night clothes, as bed sheet and it gives many more purposes*"-[FGD with women HDA].

This problem is further worsened by the perception that the bed net is not used for more than two years as to the report from most individual interviews and FGD.

## Discussion

In this study, we have explored the perception of parents of SAC in the Kutcha district on the cause of malaria and their experience of malaria prevention as well as the challenges of malaria prevention. There are multiple and various perceptions on the cause of malaria, including an explanation of the right cause and its transmission through the bite of mosquitoes. Malaria prevention experience of parents was highly influenced by their perceived causes. The challenges identified for the control are some negative attitudes towards malaria prevention strategies, the behavior of the children and the use of bed nets for unintended purposes. There was hardly any literature on the perception of parents of SAC and the comparisons and discussions made here are irrespective of the different segments of a given population.

Surprisingly, perceived cause of malaria reported by parents of SAC is hunger and drought. Exposure of hunger and shortage of food in quantity and its diversity for longer time might reduce their immunity at the individual level as well as herd immunity to protect them from malaria and increased their susceptibility to malaria infections. The other possible explanation for draught association could be giving more emphasis for securing food and ignoring malaria prevention measures during the drought period. This finding is corroborated by a mixed method study conducted in Jima Zone [29]. However, it is in contrast to the findings of studies undertaken in Konso [28] and Benishangul Gumz [34] in Ethiopia and Gabon [30], Ghana [35] and Laos [36] outside Ethiopia. The possible reason for this might be difference in the level of food insecurity in the areas of these studies [37] or difference in educational status of the participants.

The second important perceived cause of malaria is poor personal and environmental hygiene as well as consumption of contaminated food or water. This perception may be the impact of different stakeholders working to improve sanitation and hygiene, or it may be misunderstanding of the taught on management of the environment to avoid pooling of water around the compound to prevent the breeding of mosquito vectors which further implies poor awareness creation for informed decision by the community to participate in combating malaria. This finding was in line with results from Konso [28], Jima [29], Mandura district in Benishangul-Gumz region [34], an article published in 1992 among Adangibe farming community in Ghana [38, 39] and Lereto area in Peru [31] but not with findings in Gabon [30], Kersa district in eastern Ethiopia [26], and Ghana [35]. These disagreements could be due to differences in the awareness among the study population as those in Kersa district are exposed more on different researches since it has been serving as Health and demographic research site for Haromaya University. The other possible explanation might be a difference in the data collection technique used and the wider education campaign by ministry of education in Timor-leste [23].

The other cause of malaria reported by the participants was ingestion of some sweet food types. Ingestion of unripe or sweet foods like mango or sugar is also perceived as the cause of

malaria in addition to mosquito bites in Southern Ghana by two earlier studies [38, 39]. Unlike these earlier studies, recent ones conducted in Ethiopia [28, 29, 34] and other countries [30, 35, 36] did not associate malaria with these sweet and unripe food types. This is an important alarm for malaria control programs at a national and local level to increase the awareness of the community for effective implementation and well informed participation in the different malaria control strategies. Their perceived association of ingestion of unripe maize might be due to coinciding time of flowering of maize and a high transmission season where mosquito bread in high number.

The other important perceived cause of malaria as reported in studies from Konso and Ghana, unlike those in Jima [29], Benishangul-Gumz [34], Kersa [26], a recent study in Ghana [35] and Laos [36] was exposure to hot sunshine or heavy activities in such hot environmental condition. It emanates from their misunderstanding that malaria exists in an individual and will be stimulated to progress to a disease state in exposure to such hot environment as belief of most people living in Loreto area in Peru [31].The relationship between hot sunshine and malaria could be related to the biology of the vector involved in the transmission of malaria. The vector develops into adults; take blood from diseased individual and transmission to the next person follows after the rainy when there is slightly hotter sunshine.

The prevention measures that the parents of SAC experience in Kutch district were diverse and strongly influenced by their perception on the causes of malaria. People who are affected get their treatment from homemade remedies or visit health facilities for formal anti-malarial drugs. The homemade remedies are diverse and that diversity in different research might be due to differences in the herbs grown in an area [35]. Homemade remedies might be taken because of economic problems to fund as some use traditional healers for treatment [36]. Parents involved in the study have shown a positive attitude to visit the health facilities which are influenced by their satisfaction with the health service in their previous visit [28, 36].

The prevention of malaria in the kutcha district is highly challenged by the attitude of people towards the bed nets and on their perception of the causes of malaria. As a result bed nets are used for different purposes than their intended use. The unintended use of bed nets were partly due to low socioeconomic status like using the bed nets for bed sheet, head scarf and curtains [27] while others are due to their bad attitude towards the bed nets [21, 23, 35]. Due to less awareness on the importance of bed nets, people give away their own bed nets to other people in Adami Tulu in Ethiopia [26, 40, 41].

There were certain limitations and strengths associated with this study. One of the limitations is that the study area is limited to one district. The strengths of this study were involving diverse population for the study that enabled us to triangulate the information and collection of data from large number of individuals.

## Conclusion and recommendations

Parents of SAC perceive that there were multiple causes for malaria, including the bite of an infected mosquito. The misconceptions reported as the causes of malaria are poor personal and environmental hygiene, ingestion of contaminated food or water, exposure to hot environmental condition, hunger or failure to take food and consuming some sweet food items and unripe maize. Their malaria prevention experience was highly influenced by the perceived causes of malaria even though they have a good attitude for the diagnosis and treatment from health facility, environmental management and bed nets for vector control. However, effective and consistent utilization of bed net is bottlenecked by a bad attitude towards bed nets and its use for unintended purposes. Since the perception of cause of malaria determines their malaria prevention experiences, strong awareness creation to make well-informed decisions for the

participation in the malaria control strategies is needed to achieve the national malaria control targets and the vision to see the world free from malaria.

## Acknowledgments

We would like to thank Kutcha district health office for informing kebeles where malaria transmissions occur. Our gratitude also goes to participants of the research for devoting their time while providing valuable information on their lived experiences.

## Author Contributions

**Conceptualization:** Zerihun Zerdo, Jean-Pierre Van Geertruyden, Fekadu Massebo, Gelila Biresaw, Misgun Shewangizawu, Abayneh Tunje, Yilma Chisha, Tsegaye Yohanes, Hilde Bastiaens.

**Data curation:** Zerihun Zerdo, Misgun Shewangizawu.

**Formal analysis:** Zerihun Zerdo, Misgun Shewangizawu, Hilde Bastiaens, Sibyl Anthierens.

**Funding acquisition:** Fekadu Massebo, Gelila Biresaw, Abayneh Tunje, Yilma Chisha, Tsegaye Yohanes.

**Investigation:** Zerihun Zerdo.

**Methodology:** Zerihun Zerdo, Jean-Pierre Van Geertruyden, Misgun Shewangizawu, Abayneh Tunje, Hilde Bastiaens, Sibyl Anthierens.

**Supervision:** Jean-Pierre Van Geertruyden, Sibyl Anthierens.

**Writing – original draft:** Zerihun Zerdo, Jean-Pierre Van Geertruyden.

**Writing – review & editing:** Zerihun Zerdo, Jean-Pierre Van Geertruyden, Hilde Bastiaens, Sibyl Anthierens.

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
