## [Decision Letter · Decision Letter 0]

12 Feb 2020

PONE-D-19-26659

Parents’ perception on cause of malaria and their malaria prevention experience among school-aged children in Kutcha district, Southern Ethiopia; Qualitative study

PLOS ONE

Dear Mr Zeleke,

Thank you for submitting your manuscript to PLOS ONE. After careful consideration, we feel that it has merit but does not fully meet PLOS ONE’s publication criteria as it currently stands. Therefore, we invite you to submit a revised version of the manuscript that addresses the points raised during the review process.

We would appreciate receiving your revised manuscript by Mar 28 2020 11:59PM. To enhance the reproducibility of your results, we recommend that if applicable you deposit your laboratory protocols in protocols.io, where a protocol can be assigned its own identifier (DOI) such that it can be cited independently in the future. For instructions see: http://journals.plos.org/plosone/s/submission-guidelines#loc-laboratory-protocols

We look forward to receiving your revised manuscript.

Kind regards,

Stella Babalola, Ph.D

Academic Editor

PLOS ONE

Journal Requirements:

1. Please specify in your ethics statement whether participant consent was written or verbal. If verbal, please also specify: a) whether the ethics committee approved the verbal consent procedure, b) why written consent could not be obtained, and c) how verbal consent was recorded.

We also ask that you please provide in your methods section, the ethics committee approval number for your study.

2. Please ensure that you refer to Figures 1 & 2 in your text as, if accepted, production will need this reference to link the reader to the figure.

3. We note that Figure [2] includes an image of a [patient / participant / in the study]. 

Reviewers' comments:

Reviewer's Responses to Questions

**Comments to the Author**

1. Is the manuscript technically sound, and do the data support the conclusions?

Reviewer #1: Yes

Reviewer #2: Yes

2. Has the statistical analysis been performed appropriately and rigorously? 

Reviewer #1: N/A

Reviewer #2: N/A

3. Have the authors made all data underlying the findings in their manuscript fully available?

Reviewer #1: No

Reviewer #2: No

4. Is the manuscript presented in an intelligible fashion and written in standard English?

Reviewer #1: Yes

Reviewer #2: No

5. Review Comments to the Author

Reviewer #1: I will like thank the editor for offering me the opportunity to review this manuscript.

This manuscript claims that though a lot has been reported on the perception of malaria in pregnant women and parents of children under five years, not much has been report on the perceptions of parents on malaria, their experiences and challenges of malaria in school age children. This is a very important report as it contribute to filling an information gap require to fully engage malaria elimination. However, as written the manuscript requires some major revision and English proofing.

1. It is very difficult to read certain sections of the manuscript.

2. The author’s claims are properly placed within the context of the available literature. Since authors are considering the malaria perceptions and prevention experiences of parents, I urge the authors to include on what parents perceive as impact of malaria on school attendance in the study area.

3. Though the data as presented supports the authors' claim, there is need to include a section that reports the perceived impact of malaria on school attendance as mentioned above in the study area. If not, the manuscript would simply be a reflection of parents’ perception of malaria in children not SAC.

4. I do not think this protocol needs to be published online as it is not new.

5. The authors indicated that the study was initially part of a randomised clinical trial, there is not trial registration number indicated.

6. The authors do not state which software was used to analyse the data. This make it difficult to be able to repeat the experiment elsewhere.

7. This study was part of school-based bed net distribution and malaria prevention education. I will like to know whether s perceptions of school teachers who are also teachers was assessed. If this was done, it is important to know their perceptions. secondly, since it is school-base distribution of net, is the malaria education targeting the children and teachers or parents or both. If the education component is targeting schools, the perception would be important.

8. If the manuscript is revise, the content could be accessible to non-specialists.

Reviewer #2: Thanks for the opportunity to review this paper. The following are my comments:

1. the paper is worth publishing if the authors will employ the services of an editor to proof read the work carefully for them. As iit is, there are too many petty grammatical errors that makes it unpleasant reading through.

2.In the abstract, the first sentence under "result" is not needed and must be deleted.

3. The authors must provide the IRB clearance number for the study

Under the result section, lines 181 - 184 should be moved to the methods/design section of the paper.

The results must be properly presented to make it pleasant and easy to read.The quotations should be separated from the descriptives, they must be made to stand alone. Also, the source of all quotations must be indicated (e.g. Female caretaker, FGD, Kutcha) etc.

4. Lines 205 - 208 is not part of the quotation and must therefore be separated and not presented in italics. This must be done throughout the result sections.

5. The conclusion is too long and must be reduced. No need to repeat most of the results in the conclusion.

The pictures/figures are poor in quality and it is either the authors will provide a more versions at the end of the paper, one per page, else, it should be removed.

6. PLOS authors have the option to publish the peer review history of their article (what does this mean?). If published, this will include your full peer review and any attached files.

Reviewer #1: No

Reviewer #2: Yes: Collins Stephen Ahorlu

---

## [Author Response · Author response to Decision Letter 0]

23 Mar 2020

Dear reviewers

I would like to appreciate giving time to give us your valuable comments to improve the quality of the manuscript. Here, I presented responses to your comments

Response to Reviewer #1

This manuscript claims that though a lot has been reported on the perception of malaria in pregnant women and parents of children under five years, not much has been report on the perceptions of parents on malaria, their experiences and challenges of malaria in school age children. This is a very important report as it contributes to filling an information gap requires to fully engage malaria elimination. However, as written the manuscript requires some major revision and English proofing.

Comment 1: It is very difficult to read certain sections of the manuscript. 

Response 1: I think this is also associated with language edition. The revised manuscript with track changes is edited for its English language which should have increased the readability.

Comment 2: The author’s claims are properly placed within the context of the available literature. Since authors are considering the malaria perceptions and prevention experiences of parents, I urge the authors to include on what parents perceive as impact of malaria on school attendance in the study area.

Response 2: The previous literature focused on pregnant mothers and parents/ mother of children under five years of age as you have also appreciated in your introduction of the comments. Parents perception on the susceptibility children varies as indicated in the result section of this study as they are least prioritized and this will have impact on their experience of malaria prevention among school-aged children. Since this study focused on the perception on cause of malaria and malaria prevention experiences of parents, we did not focused on their perception of the impact of malaria on school attendance. However, one of the key informants (school headmaster) raised the impact of malaria on school attendance: 

“When the school-aged children become fibril, people use health facility. Even if the children get diseased at the school level but we also take them to the health center. However, some of the children get febrile and go to home and may not come to the school. In such conditions, we are asking parents or write letter to come to the school and give what the condition is for the students” [school headmaster]

Since impact of malaria on school attendance is important, it should be addressed in future research.

Comment 3: Though the data as presented supports the authors' claim, there is need to include a section that reports the perceived impact of malaria on school attendance as mentioned above in the study area. If not, the manuscript would simply be a reflection of parents’ perception of malaria in children not SAC. 

Response 3: During the interview the questions are specifically focused on School-aged children. We added a table with key interview questions to clarify this in the revised manuscript with track changes. If this is more appropriate, the tables could also be presented in the appendix. 

Comment 4: I do not think this protocol needs to be published online as it is not new.

Response 4: we do not understand this comment. In his/her summary, the reviewer says it is an important report. If it is for the protocol (we understand it as proposal), it is good and appreciate the comment of the reviewer.

Comment 5: The authors indicated that the study was initially part of a randomized clinical trial, there is not trial registration number indicated.

Response 5: This is base to develop the malaria prevention education part of the trail thought to be conducted. The trail is modified and registered in Pan African Clinical Trails Registry with trial ID of PACTR202001837195738Comment 6: The authors do not state which software was used to analyze the data. This makes it difficult to be able to repeat the experiment elsewhere.

Response 6: Qualitative researches are flexible and will vary according to the context of the study area. Thus, if similar study is conducted somewhere else, it will be different. The aim of qualitative research is to be able to transfer findings to other settings, meaning sufficient information needs to be available. Dependability refers to the transparency and auditability of the research process, so being clear about the decisions you took, about how you did your research. So, we have tried to describe the procedures followed for the data collection and analysis in detail in the methods section of the revised manuscript with track changes. Thus, the readers can get good idea of the whole research process. We acknowledge that software can support the traceability of the coding process; however this was not used in our study because the analyst had no analytical software at the time of analyzing the transcript.

Comment 7: This study was part of school-based bed net distribution and malaria prevention education. I will like to know whether s perceptions of school teachers who are also teachers were assessed. If this was done, it is important to know their perceptions. Secondly, since it is school-base distribution of net, is the malaria education targeting the children and teachers or parents or both? If the education component is targeting schools, the perception would be important.

Response 7: We did not assess the perception of teachers on malaria in this study. We agree with the reviewer that this could have had an added value. However, a teacher was involved as a key informant. The intervention targeted both children and parents. The research team trained two science teachers and school headmaster from each of intervention schools. Then these trained teachers trained the children and parents. 

Comment 8: If the manuscript is revise, the content could be accessible to non-specialists.

Response 8: We could not understand what the reviewer intends to state but if our understanding is correct? We understood it as the content is good if it is revised. We have tried to address the comments of the reviewers and the editor to make readable for anyone including non-specialists.

Response to Reviewer #2

Comment 1: the paper is worth publishing if the authors will employ the services of an editor to proof read the work carefully for them. As it is, there are too many petty grammatical errors that makes it unpleasant reading through.

Response 1: it is edited for English language and hope it will be easier to read

Comment 2: In the abstract, the first sentence under "result" is not needed and must be deleted.

Response 2: the first sentence in the result section is deleted in the revised manuscript with track changes

Comment 3: 

a. The authors must provide the IRB clearance number for the study

Under 

Response: The IRB clearance number (ሕጠሳኮ/11275/21) is placed in line number 161 of the revised manuscript with track changes.

b. The result section, lines 181 - 184 should be moved to the methods/design section of the paper.

Response: yes, you are right. According to your recommendation, we have placed it in the methods section and rearrangements are made without losing the main information 

c. The results must be properly presented to make it pleasant and easy to read. The quotations should be separated from the descriptive; they must be made to stand alone. Also, the source of all quotations must be indicated (e.g. Female caretaker, FGD, Kutcha) etc.

Response: yes, you are right, we have placed quotations separately from the other parts and stand alone and the source is indicated in parenthesis next to each quotation.

Comment 4: Lines 205 - 208 is not part of the quotation and must therefore be separated and not presented in italics. This must be done throughout the result sections.

Response 4: yes, you are definitely correct and in the revised manuscript with track changes, it is outside the quotation mark and not written in italics.

Comment 5: The conclusion is too long and must be reduced. No need to repeat most of the results in the conclusion.

Response 5: The conclusion section is shortened in the revised manuscript with track change.

The pictures/figures are poor in quality and it is either the authors will provide a more versions at the end of the paper, one per page, else, it should be removed.

Response: we have tried to place each of the pictures as supporting information at the end of the revised manuscript with track changes

---

## [Editor Report · Decision Letter 1]

25 Jun 2020

PONE-D-19-26659R1

Parents’ perception on cause of malaria and their malaria prevention experience among school-aged children in Kutcha district, Southern Ethiopia; Qualitative study

PLOS ONE

Dear Dr. Zeleke,

Thank you for submitting your manuscript to PLOS ONE. After careful consideration, we feel that it has merit but does not fully meet PLOS ONE’s publication criteria as it currently stands. Therefore, we invite you to submit a revised version of the manuscript that addresses the points raised during the review process.

**The manuscript still very much needs to be copyedited carefully by either a native-English speaking colleague or a professional copy-editing service. As noted in my message earlier today, you may approach any qualified individual or any professional scientific editing service of your choice. To provide authors with an additional option, PLOS has partnered with American Journal Experts (AJE) to provide discounted services to PLOS authors. AJE has extensive experience helping authors meet PLOS guidelines and can provide language editing, translation, manuscript formatting, and figure formatting to ensure your manuscript meets our submission guidelines. If the PLOS editorial team finds any language issues in text that AJE has edited, AJE will re-edit the text for free. To take advantage of this special partnership, use the following link: https://www.aje.com/go/plos/.**

**As the Revision 1 manuscript has not been sent for external review, please could you upload the Response to Reviewers letter that responds to the comments from Reviewers #1 and #2 from the first round of reviews, when submitting your revised manuscript.**

We look forward to receiving your revised manuscript.

Kind regards,

R Matthew Chico, MPH, PhD

Academic Editor

PLOS ONE

---

## [Author Response · Author response to Decision Letter 1]

17 Jul 2020

Response to Reviewers

Dear editor and reviewers

We would like to thank the editor and the reviewers for their valuable comments and suggestions they have presented to improve the quality of our research before publication in PLOS ONE Journal.

Editor’s comment (Ed. comment)

 Ed. Comment 1: please specify in your ethics statement whether participant consent was written or verbal. If verbal, please also specify: a) whether the ethics committee approved the verbal consent procedure, b) why written consent could not be obtained, and c) how verbal consent was recorded. We also ask that you please provide in your methods section, the ethics committee approval number for your study

Response to Ed. comment 1: This study is approved with verbal consent procedure to be followed by the Institutional Research Ethics review Board (IRB) in College of Medicine and Health Sciences, Arba Minch University with the reference number of ሕጠሳኮ/11275/21. Verbal consent is obtained from each participant after detailed information on purpose, procedure, benefit or harms and autonomy to participate is given. Verbal consent is preferred because majority of the residents in the study area were illiterate, no biological specimen is collected or sensitive issue is discussed. The verbal consent obtained from each participant in the study is documented in college of Medicine and Health Sciences at Arba Minch University. This is indicated in the manuscript with track changes.

Ed. Comment 2. Please ensure that you refer to Figures 1 & 2 in your text as, if accepted, production will need this reference to link the reader to the figure.

Response to Ed. Comment 2: figure 1 refers to the occupational status of the participants in the study and it is referenced in line 206 and 207 in the revised manuscript with track changes when viewed in final form of review. Figure 2 is referred in the main text depending on what the specific pictures in the figure refer to at different places (in line number 185, 326, 360 and 361) in the revised manuscript with track changes (when viewed in final form of review). 

Ed. Comment 3. We note that Figure [2] includes an image of a [patient / participant / in the study]. 

Response to Ed. Comment 3: Figure 2 indeed contains an image of a person and he is indicating as bed net is used to make rope that would be used to handle cattle. The individual in figure 2 has provided consent for publication of this picture. 

Response to Reviewer #1

This manuscript claims that though a lot has been reported on the perception of malaria in pregnant women and parents of children under five years, not much has been report on the perceptions of parents on malaria, their experiences and challenges of malaria in school age children. This is a very important report as it contributes to filling an information gap requires to fully engage malaria elimination. However, as written the manuscript requires some major revision and English proofing.

Comment 1: It is very difficult to read certain sections of the manuscript. 

Response 1: I think this is also associated with language edition. The revised manuscript with track changes is edited for its English language by professor from USA which should have increased the readability.

Comment 2: The author’s claims are properly placed within the context of the available literature. Since authors are considering the malaria perceptions and prevention experiences of parents, I urge the authors to include on what parents perceive as impact of malaria on school attendance in the study area.

Response 2: The previous literature focused on pregnant mothers and parents/ mother of children under five years of age as you have also appreciated in your introduction of the comments. Parents perception on the susceptibility of children varies as indicated in the result section of this study as they are least prioritized and this will have impact on their experience of malaria prevention among school-aged children. Since this study focused on the perception on cause of malaria and malaria prevention experiences of parents, we did not focused on their perception of the impact of malaria on school attendance. However, one of the key informants (school headmaster) raised the impact of malaria on school attendance: 

“When the school-aged children become fibril, people use health facility. Even if the children get diseased at the school level but we also take them to the health center. However, some of the children get febrile and go to home and may not come to the school. In such conditions, we are asking parents or write letter to come to the school and give what the condition is for the students” [school headmaster]

Since impact of malaria on school attendance is important, it should be addressed in future research.

Comment 3: Though the data as presented supports the authors' claim, there is need to include a section that reports the perceived impact of malaria on school attendance as mentioned above in the study area. If not, the manuscript would simply be a reflection of parents’ perception of malaria in children not SAC. 

Response 3: During the interview the questions are specifically focused on School-aged children. We added a table with key interview questions to clarify this in the revised manuscript with track changes. If this is more appropriate, the tables could also be presented in the appendix. 

Comment 4: I do not think this protocol needs to be published online as it is not new.

Response 4: we do not understand this comment. In his/her summary, the reviewer says it is an important report. If it is for the protocol (we understand it as proposal), it is good and appreciate the comment of the reviewer.

Comment 5: The authors indicated that the study was initially part of a randomized clinical trial, there is not trial registration number indicated.

Response 5: This is base to develop the malaria prevention education part of the trail thought to be conducted. The trail is modified and registered in Pan African Clinical Trails Registry with trial ID of PACTR202001837195738

Comment 6: The authors do not state which software was used to analyze the data. This makes it difficult to be able to repeat the experiment elsewhere.

Response 6: Qualitative researches are flexible and will vary according to the context of the study area. Thus, if similar study is conducted somewhere else, it will be different. The aim of qualitative research is to be able to transfer findings to other settings, meaning sufficient information needs to be available. Dependability refers to the transparency and auditability of the research process, so being clear about the decisions you took, about how you did your research. So, we have tried to describe the procedures followed for the data collection and analysis in detail in the methods section of the revised manuscript with track changes. Thus, the readers can get good idea of the whole research process. We acknowledge that software can support the traceability of the coding process; however this was not used in our study because the analyst had no analytical software at the time of analyzing the transcript.

Comment 7: This study was part of school-based bed net distribution and malaria prevention education. I will like to know whether perceptions of school teachers who are also teachers were assessed. If this was done, it is important to know their perceptions. Secondly, since it is school-base distribution of net, is the malaria education targeting the children and teachers or parents or both? If the education component is targeting schools, the perception would be important.

Response 7: We did not assess the perception of teachers on malaria in this study. We agree with the reviewer that this could have had an added value. However, a teacher was involved as a key informant. The intervention targeted both children and parents. The research team trained two science teachers and school headmaster from each of intervention schools. Then these trained teachers trained the children and parents. The process is monitored for its doze, reach, fidelity and the reelection of trainers, trainees on the training by independent monitors.

Comment 8: If the manuscript is revising, the content could be accessible to non-specialists.

Response 8: We could not understand what the reviewer intends to state but if our understanding is correct? We understood it as the content is good if it is revised. We have tried to address the comments of the reviewers and the editor to make readable for anyone including non-specialists through language edition by a professor from USA.

Response to Reviewer #2

Comment 1: the paper is worth publishing if the authors will employ the services of an editor to proof read the work carefully for them. As it is, there are too many petty grammatical errors that makes it unpleasant reading through.

Response 1: it is edited for English language by professor from USA and hope it will be easier to read

Comment 2: In the abstract, the first sentence under "result" is not needed and must be deleted.

Response 2: the first sentence in the result section is deleted in the revised manuscript with track changes

Comment 3: 

a. The authors must provide the IRB clearance number for the study

Under 

Response: The IRB clearance number (ሕጠሳኮ/11275/21) is placed in line number 178 of the revised manuscript with track changes (when viewed in final form of review).

b. The result section, lines 181 - 184 should be moved to the methods/design section of the paper.

Response: yes, you are right. According to your recommendation, we have placed it in the methods section and rearrangements are made without losing the main information 

c. The results must be properly presented to make it pleasant and easy to read. The quotations should be separated from the descriptive; they must be made to stand alone. Also, the source of all quotations must be indicated (e.g. Female caretaker, FGD, Kutcha) etc.

Response: yes, you are right, we have placed quotations separately from the other parts and stand alone and the source is indicated in parenthesis next to each quotation.

Comment 4: Lines 205 - 208 is not part of the quotation and must therefore be separated and not presented in italics. This must be done throughout the result sections.

Response 4: yes, you are definitely correct and in the revised manuscript with track changes, it is outside the quotation mark and not written in italics.

Comment 5: The conclusion is too long and must be reduced. No need to repeat most of the results in the conclusion.

Response 5: The conclusion section is shortened in the revised manuscript with track change.

The pictures/figures are poor in quality and it is either the authors will provide a more versions at the end of the paper, one per page, else, it should be removed.

Response: we have tried to place each of the pictures as supporting information at the end of the revised manuscript with track changes

---

## [Decision Letter · Decision Letter 2]

14 Sep 2020

Parents’ perception on cause of malaria and their malaria prevention experience among school-aged children in Kutcha district, Southern Ethiopia; Qualitative study

PONE-D-19-26659R2

Dear Dr. Zeleke,

We’re pleased to inform you that your manuscript has been judged scientifically suitable for publication and will be formally accepted for publication once it meets all outstanding technical requirements.

Kind regards,

Gabriel A. Picone

Academic Editor

PLOS ONE

Additional Editor Comments (optional):

Reviewers' comments:

Reviewer's Responses to Questions

**Comments to the Author**

1. If the authors have adequately addressed your comments raised in a previous round of review and you feel that this manuscript is now acceptable for publication, you may indicate that here to bypass the “Comments to the Author” section, enter your conflict of interest statement in the “Confidential to Editor” section, and submit your "Accept" recommendation.

Reviewer #1: All comments have been addressed

2. Is the manuscript technically sound, and do the data support the conclusions?

Reviewer #1: Yes

3. Has the statistical analysis been performed appropriately and rigorously? 

Reviewer #1: N/A

4. Have the authors made all data underlying the findings in their manuscript fully available?

Reviewer #1: Yes

5. Is the manuscript presented in an intelligible fashion and written in standard English?

Reviewer #1: Yes

6. Review Comments to the Author

Reviewer #1: The authors have responded to all the comments and the manuscript has been greatly improved and can be accepted for publication.

7. PLOS authors have the option to publish the peer review history of their article (what does this mean?). If published, this will include your full peer review and any attached files.

Reviewer #1: **Yes: **Dr. Ndong Ignatius Cheng,

Senior Lecturer,

Department of Biochemistry,

Catholic University of Cameroon (CATUC) Bamenda

Research Fellow

Department of Epidemiology,

Noguchi Memorial Institute for Medical Research,

College of Health Sciences, University of Ghana

---

## [Editor Report · Acceptance letter]

1 Oct 2020

PONE-D-19-26659R2 

Parents’ perception on cause of malaria and their malaria prevention experience among school-aged children in Kutcha district, Southern Ethiopia; Qualitative study 

Dear Dr. Zerdo:

I'm pleased to inform you that your manuscript has been deemed suitable for publication in PLOS ONE. Congratulations! Your manuscript is now with our production department. 

Kind regards, 

on behalf of

Dr. Gabriel A. Picone 

Academic Editor

PLOS ONE